# Positioning Chaplaincy in the Pluralistic and Multidisciplinary Dutch Care Context

**Anja Visser \*** , **Hetty Zock and Hanneke Muthert**

Faculty of Religion, Culture and Society, University of Groningen, 9712 GK Groningen, The Netherlands
\* Correspondence: a.visser-nieraeth@rug.nl

**Abstract:** The professional identity of chaplains is under question because of societal trends of disaffiliation from and pluralization of religion, and of deinstitutionalization of care. Chaplaincy in the Netherlands looks to discourse around "meaning" to navigate these challenges. The use of the term "meaning" as the central concept in the professional identity of chaplaincy (and, by extension, spiritual care) is not undisputed, however. There are three related critiques: 1. Meaning and meaning-making have a strong cognitive and intentional connotation, which does not do justice to the lived experience of meaning and might lead to a medicalization of meaning. 2. The term meaning places the professional identity of chaplaincy in the instrumental discourse of other professions, which might lead to "abuse" of spiritual care toward external objectives such as health, (hedonistic) well-being, and/or economic gain, instead of internal objectives such as faith and spirituality. 3. A focus on meaning leads to a marginalization of religion, both societally and within chaplaincy, which might negatively affect chaplaincy's core competence of hermeneutic understanding and worldview counseling. We conclude that finding one language to present the discipline might not be feasible and desirable. Instead, we advocate for the revitalization of the hermeneutic competency of chaplains.

**Keywords:** chaplaincy; Netherlands; professional identity

## 1. Introduction

Throughout the world, and markedly in the Netherlands, the professional identity of chaplains is under question because of societal trends of disaffiliation from and pluralization of religion, and of deinstitutionalization of care.

Disaffiliation and pluralization challenge the field to find roles and tasks beyond traditional forms of spiritual care to remain visible and relevant in collaboration with other professionals. They also challenge the field to develop a language and identity that includes the plurality of worldviews, but that still conveys its professional core.

Deinstitutionalization of care and, with that, the move of chaplains in the Netherlands to primary care and the social domain have intensified the collaboration between chaplains and other professions, and have strengthened the need for other professionals to take up responsibilities in the spiritual domain of care. This challenges chaplains to also develop a language around spiritual care that helps them to show how responsibilities are entangled and how expertise and responsibilities in this domain differ between professions.

Currently, chaplaincy in the Netherlands looks to discourse around "meaning" to navigate these challenges. However, this also poses various risks to the professional identity of chaplains. From this perspective, we examine the discussion around using "meaning" as a central concept in chaplaincy.

We conclude that finding one language to present the discipline might not be feasible or desirable. Instead, we advocate for embracing the different expressions of religion, spirituality, and meaning, as well as the different ways in which this is approached by various professionals, and revitalizing the hermeneutical practice of chaplains to navigate this plurality.

## 2. A Changing Professional Identity

Professional identity is a multidimensional concept. Based on previous definitions, Ellis and Hogard (2020) suggest that professional identity in caring professions contains three main features: knowledge, competency, and values (or ethical standards). They further state that professional identity has both internal aspects—the way the profession or professional sees themselves—and external aspects—the way others see the profession or professional. However, according to Ellis and Hogard, professional identities are not directly observable; they have to be inferred from observable behavior. This is especially true for people outside the profession, but even within the profession, a majority of the knowledge, competencies, and values remains implicit, learned by practice. This was also evident in the Dutch Case Studies Project, in which chaplains were challenged to articulate the relationship between their practice and chaplaincy theory (Muthert et al. 2019). It was concluded that chaplains obtain a body of knowledge; they come to embody theory and formal professional knowledge. Decisions regarding intervention are then based on the interaction of this knowledge with interpersonal processes and cultural factors. This makes it feel like decisions are made intuitively, and it becomes difficult to articulate how professionals work and why they do so in this way.

Looking at the formal professional identity of chaplains in the Netherlands, the current professional standard (Dutch Association of Spiritual Caregivers VGVZ 2015, pp. 7–9) suggests that: "Chaplaincy is professional support, counseling and advice for meaning and belief systems". They are described as offering individual and group counseling, by which they search for what provides the client with power and inspiration, in a way that fits their life stories and worldview backgrounds. Chaplains can also support clients with rituals and sacraments. In some contexts, they support the constitutional right to freedom of religion. According to the professional standard, chaplains are also available for other professionals and for management. They provide advice in the areas of worldview and ethics, provide training, and support a "worldview climate" in institutions. According to the professional standard, chaplains have competency in three main areas: (1) content-related, in the areas of meaning-making processes and worldview, of religion and spirituality, and of ethics; (2) process-related, in terms of being able to share their knowledge and considerations with others and to have a relational function; (3) person-related, through having an authentically lived spirituality that they actively maintain and that they use as a basis for their work. As might be expected, the values or ethical standards underlying the profession remain somewhat implicit, but from the section on the quality standard several values can be deduced—for example, carefulness through focused and methodical attention, beneficence through contributing to the well-being and functioning of others and of organizations, person-centeredness, authenticity or congruence, and freedom of religion.

This professional profile or identity already reflects a shift in how chaplaincy positions itself. Most notably, the institutionally religious dimension of chaplaincy has diminished: in the 2002 professional standard, chaplaincy was defined as "The professional and ex officio support and counseling for people searching for meaning to their existence, from and based on their faith and belief system (. . .)" (VGVZ 2002, p. 8). In the 2015 definition, "faith" has been replaced with "meaning". In her discussion of the changing profile of chaplaincy, Zock (2022a, p. 43) also notes that chaplains have become "meaning-making experts" rather than religious functionaries. Additionally, no reference is made to the "office" of chaplains by the 2015 definition. Instead, more general descriptions appear, such as the statement that chaplains work on the basis of a qualification received from a "societally recognized worldview institute" or an authorization from a professionally recognized body, the RING-GV, and that the chaplain must have an "authentically lived spirituality".

The religious disaffiliation and pluralization of Dutch society are a strong force behind this change in the professional identity of chaplaincy. According to Statistics Netherlands the percentage of the Dutch population that is a member of a religious denomination has declined from 67% in 1966 to 43% in 2022. Similarly, the percentage of people visiting religious services at least once a week has declined from 50% in 1966 to 6.7% in 2022

(Schmeets and Houben 2023; Hart 2011). The proportion of people indicating that they do not believe in God has increased from 6% in 1966 to 33% in 2019 (Statistics Netherlands 2020; Hart 2011). At the same time, it is suggested that the number of religious denominations has increased dramatically. In 1966, 7% of the Dutch religious population belonged to a different religious denomination than the Roman Catholic church or the Protestant churches that formed the Dutch Protestant church in the year 2000, whereas in 2022, 21% of the Dutch population indicated that they belonged to a different religious denomination than the Roman Catholic or Dutch Protestant church (Schmeets and Houben 2023; Hart 2011). Similarly, the percentage of the Dutch population that claims to belong to a different religious denomination to Christianity has increased from 4% in 1960 to 12% in 2022 (Schmeets and Houben 2023). Research by Berghuijs (2017) indicates that about 23% of the Dutch population reports some association with multiple religions, mostly Christianity and either Buddhism or Judaism. This multiple religious belonging ranges from feeling affinity with a religion to self-identification as a member of a religion.

This disaffiliation and pluralization have had two consequences for chaplaincy: its relevance came under scrutiny and the worldview of chaplains themselves also became more pluralistic. In response to questions about their relevance, chaplains gradually expanded their roles and tasks within the institutions at which they were employed. From the 1980s onward, they began to contribute to the education of care professionals, provided moral counseling, participated in ethics committees, and began to take an advisory role toward management (Zock 2019). In response to the worldview pluralization of chaplains themselves, a "sector" of non-denominational chaplaincy was introduced in the VGVZ in 2014, along with the already existing sectors of Protestant, Catholic, Islamic, Jewish, Hindu, and Humanist chaplains (later, Orthodox and Buddhist sectors were also added). These non-denominational spiritual caregivers were either not affiliated with a denomination or did not want to work as a representative of the denomination (VGVZ 2010).

Despite having gone through all this, the professional identity of chaplaincy is again (or still) under debate in the Netherlands. This time, the continuing deinstitutionalization of care seems to be the main driving force. Deinstitutionalization of care seems to ask for increased multi- and interdisciplinary collaboration in spiritual care provision. However, such collaboration requires a clear external professional identity for all parties involved, because the various professionals need to understand each other's expertise to be able to call upon each other at the right time and for the right questions. Several studies have shown, however, that it is difficult for chaplains in the Netherlands to find and communicate a fitting external identity: Many professionals either associate chaplaincy with church ministry or do not feel that chaplains have added value for the care that the professionals themselves provide (Meurs et al. 2023).

The increased collaboration with other professionals is asked because the demand for spiritual care is expected to increase with the changing care landscape, and people needing spiritual care increasingly reside in places to which chaplains do not have easy access. Historically, chaplaincy was situated almost exclusively in institutions and in communities through religious organizations (Zock 2019). The deinstitutionalization of care means that people now reside in (mental) healthcare institutions for shorter periods of time, but with increasingly more complex complaints. This development both reduces access to spiritual care—which is mainly provided in institutions—but also suggests that the demand for spiritual care in institutions might increase. Additionally, access to spiritual care outside of institutions is reduced due to disaffiliation and secularization, because fewer people receive spiritual care facilitated by religious institutions. Thus, it is presumed that there is a growing demand for spiritual care in primary care and the social domain, as well. The latter is also evidenced by the increased societal attention to spirituality and meaning in life. All this means that care professionals in all care domains have a greater responsibility in spiritual care, through providing spiritual care themselves and referring specific cases to chaplains. This is evidenced in, for example, the inclusion of the spiritual domain in the

Dutch quality standards for palliative care and for mental health care (IKNL/Palliactief 2017; Akwa GGZ 2023).

Influenced by these developments, Dutch chaplains have started many initiatives to formalize spiritual care provision outside of institutions and to build on their external professional identities. In 2019, the Dutch ministry of health, wellness, and sports supported this movement by subsidizing spiritual care in primary care and the social domain for people aged 50 years or older and children and adults receiving palliative care. This initiative led to two websites to familiarize funding bodies, care professionals, and citizens with spiritual care and chaplaincy, and to demonstrate its added value. The number of chaplaincy organizations in primary care and the social domain has also grown exponentially, resulting in a reorganization in 2022 into 16 regional Centra voor Levensvragen (literal translation: Centres for Life Questions) and an association in which these centers collaborate (the Vereniging Samenwerkende Centra voor Levensvragen). One of the activities of the chaplains associated with the centers is providing training for care professionals in recognizing spiritual needs and in providing first support. These activities have brought about growing recognition of the relevance of spiritual care, but also a blurring of boundaries between chaplaincy and other care professions.

In an effort to more clearly convey its professional identity to care professionals and funding bodies, the term "meaning" (in Dutch: zingeving) is becoming more central in communication by chaplains and regarding chaplaincy. This term is used in the hope and expectation that it will assist in developing one language with which to speak about spiritual care in a pluralistic and multidisciplinary context. As an example of the dominance of "meaning" in current communication about chaplaincy and, more generally, spiritual care, the funding body that finances projects for the professionalization of chaplaincy in primary care and the social domain, and which is also the main funding body for health research in the Netherlands, describes its aim as "stimulating the development and implementation of meaning-making and chaplaincy (in Dutch: zingeving en geestelijke verzorging) in care and welfare" (ZonMw n.d.). A national project in this program is the Knowledge Hub Meaning-making (Kenniswerkplaats Zingeving), which describes spiritual care by other professionals than chaplains as "care for meaning".

### 3. The Debate on the Move to "Meaning"

The use of the term "meaning" as the central concept in the professional identity of chaplaincy (and, by extension, spiritual care) is not undisputed, however. This debate revolves around three related critiques:

1. Meaning and meaning-making have strong cognitive and intentional connotations, which do not do justice to the lived experience of meaning and might lead to a medicalization of meaning.
2. The term meaning places the professional identity of chaplaincy in the instrumental discourse of other professions, which might lead to "abuse" of spiritual care toward external objectives such as health and (hedonistic) well-being and/or economic gain, instead of internal objectives such as faith and spirituality.
3. A focus on meaning leads to a marginalization of religion, both societally and within chaplaincy, which might negatively affect their core competence of hermeneutic understanding and worldview counseling.

We will discuss each of these critiques in turn.

### 3.1. Meaning as Cognitive and Intentional

The terms meaning and meaning-making are often used in various disciplines. More often than not, they refer to individual experiences of understanding and significance (or purpose): How do I understand life and myself, and what is of importance or value in my life (or, what is worth pursuing)? In psychological terms, these aspects refer to appraisal and goal-orientation. Various techniques, such as cognitive-behavioral therapy, have been developed to influence this type of meaning-making. Chaplains note that if this is how the

term "meaning" is understood, it runs the risk of being viewed as something that can be controlled and, therefore, made part of treatment or "medicalized".

Instead, chaplains draw attention to terms such as "existential meaning" to signify that meaning-making is more fundamental than self-understanding, understanding of life events, and finding valuable goals to pursue (Mooren and Walton 2013). Such forms of meaning also go beyond conscious thought and control. In this regard, Alma (2020, pp. 21–41) discusses the physical and unintentional nature of meaning. She places the need for meaning in the context of what we consider to be good. When we find this, it makes us feel enriched, fulfilled, and "at home", with a desire to engage with life. However, as Alma illustrates, we cannot predict when we will find this sense of meaning or when we will lose it; it can happen all of a sudden due to something we see, hear, feel, taste, or smell, and we feel it as shocks in our bodies and/or as emotions. Only in second instance do we reflect on this and try to find words to understand and express our experiences. Alma, therefore, introduces the term "exploration of life" or "life orientation" ("levensverkenning") to express the experimental, searching, and continuous orientation to the good life, in other words, to meaning (Alma 2020, pp. 83–121).

Reinforcing the idea that meaning-making is, perhaps most of the time, not an intentional process, various authors stress the importance of imagination for meaning-making. They consider imagination—and with that, meaning-making—relational processes. Imagination requires that a person allow different perspectives to "speak to" or "resonate with" their existing perspectives. This dialogical process makes it possible to find a different meaning, or for existing meanings to become more explicit and deeply understood (Alma 2020). Different perspectives can be encountered anywhere, in (religious) writing, music, dance, visual arts, silence, and conversation. This viewpoint also implies that meaning is found in the world and not necessarily in the individual, as the dominant understanding of meaning seems to be. Of course, this understanding of meaning does not completely preclude its medicalization, given that chaplains and different types of therapists intentionally engage the imaginations of their clients to influence the search for meaning. It does, however, show limitations to the idea that people "make" meaning.

### 3.2. Meaning as Instrumental to External Objectives

Meaning is often related to well-being and quality of life. Some chaplains worry, therefore, that adopting this term as central to their professional identity will result in an instrumentalization of this discipline toward objectives such as productivity, societal participation, and cost reduction. In a sense, this trend can already be observed in studies that examine the effects of chaplaincy on well-being, (mental) health, and care costs (Alghanim et al. 2021; Buelens et al. 2023; Kirchoff et al. 2021). Molenaar (2016, 2022) also notes this movement in the professional profile by the VGVZ, where it states that: "Through the systematic and methodical attention to meaning and belief systems, spiritual caregivers contribute to the well-being of people in relation to themselves, others and their environment." (VGVZ 2015, p. 9). According to Molenaar, such instrumentalization of chaplaincy for well-being introduces utilitarian ethics into spiritual care, whereas chaplaincy should primarily be founded in care ethics and spirituality (Molenaar 2022). Utilitarianism suggests that the right thing to do is that which brings happiness and pleasure to the largest amount of people (West and Duignan 2023). On the other hand, care ethics as applied in the Netherlands suggests that care professionals should work from a relational attitude, in which they strive for the maintenance and flourishing of relationships. It focuses on values such as commitment, dependence, vulnerability, responsibility, and caring (Vries 2016). Such a relational attitude might not bring about happiness or pleasure, but might also mean enduring unhappiness and pain together. The experience of relationship and the acknowledgement of pain and suffering are also fundamental to spirituality.

This concern might be a legitimate critique of the current neo-liberal and utilitarian thinking in Dutch society, to which chaplains have to relate. However, at present, there seems to be little cause for worry about this dominating the professional identity of chap-

laincy. By looking at descriptions of goals of chaplaincy in the Netherlands in literature published between 2014 and 2021, Visser et al. (2023) concluded that most authors position these goals in the domain of worldview/life view and spirituality, and not in the domain of well-being (though some also do). In a current study, we are examining the nature of chaplaincy through interviews with clients of chaplains and through focus groups with chaplains, other professionals, and chaplaincy clients. Preliminary results indicate that they describe chaplaincy as operating on four dimensions: worldview development, coping with life events and circumstances, relational affirmation, and experiences of connectedness. Clients particularly stressed the importance of relational affirmation in the experienced benefits of chaplaincy. In descriptions of outcomes of the chaplaincy encounter, clients sometimes mention aspects related to mental health and well-being that are not hedonistic, as promoted in utilitarianism (such as feeling calm, being at peace, or feeling resilience). However, most often, the experiences relate to being more aware of what is happening in the present moment (for example, feeling space within oneself, feeling seen and heard, being able to articulate words and feelings, or being better able to endure circumstances).

It might also be practically disadvantageous for chaplaincy to present itself as a profession that contributes to health and hedonistic or emotional well-being or quality of life. This is because, empirically, there is little evidence that spirituality or spiritual care contribute substantially to these outcomes. Studies on the relationship between spirituality, spiritual care, emotional well-being, and quality of life consistently show very small effect sizes that rarely reach statistical significance (Buelens et al. 2023; Garssen et al. 2021; Kirchoff et al. 2021). Alghanim et al. (2021) even found that introducing chaplains to an intensive care unit (ICU) who facilitated communication, offered support, and organized multidisciplinary family-meetings was associated with longer stays in the hospital and the ICU, and with higher costs compared to a control group that did not work with a chaplain. This does not mean that chaplaincy might not indirectly affect well-being and quality of life. However, it does mean that "well-being"—and with that, meaning—might not be the best fit to convey its more direct expertise and effects.

### 3.3. Meaning as Marginalization of Religion

A final concern around using meaning as a central concept in chaplaincy is that it is associated with psychological and humanistic discourse, and less with religious discourse. In their study around the goals of chaplaincy, Visser et al. (2023) also noted that the language used to describe the goals of chaplaincy was quite psychological, humanistic, or secular; rarely did references to specific religious concepts appear in the texts. We also noted this above with regard to the professional standard: references to life view and faith, and to the office of the chaplain, are no longer included in the 2015 definition of chaplaincy. A marginalization of religion in chaplaincy can also be deduced from the observation that the sector for non-denominational chaplains is the fastest-growing sector of the VGVZ. These developments point to a strategy of "neutralizing". In the American context, Cadge and Sigalow (2013) define neutralizing as the use of "a broad language of spirituality that emphasizes commonalities rather than differences". For the Dutch context, we would replace the word "spirituality" with "meaning".

A marginalization of religion within chaplaincy raises questions about the spiritual identity of chaplaincy. Smeets and Morice-Calkhoven (2014) have suggested that the spiritual identity (or "spiritual competence", as it is called by the authors) can be viewed on two levels: (a) internally, when it concerns the extent to which chaplains have their own lived worldview and are able to identify and interact with the worldviews of others; and (b) externally, when it concerns the differentiation between chaplaincy and other care professions, and the societal legitimation of the profession.

When looking at the internal spiritual identity, there seems to be little cause for concern. Studies among chaplains suggest that the spiritual/religious orientation of the chaplain still plays a significant role in spiritual care encounters (Toom 2022; Liefbroer and Olsman 2020; Koelewijn-Wiersma 2023). Institutionally, various measures have also been

taken to ensure this competence. In Section 2, we mentioned that the professional standard contains a description of content-related and process-related competences that represent the internal spiritual identity. Also, chaplains need a qualification from a "societally recognized worldview institute" or authorization from a professionally recognized body, the RING-GV, that attests to their internal spiritual identity before being admitted to the Dutch quality register for chaplains (SKGV). Finally, master's programs for non-denominational chaplains, such as our own, include learning lines in which students learn to articulate their own worldview.

It seems to be mainly the external spiritual identity that is affected by marginalization of religion. In Section 2, we mentioned that studies have shown that professionals in primary care and the social domain regularly question the relevance of chaplaincy, because they feel that they already provide sufficient spiritual care themselves. This seems to happen especially when spiritual care is presented as "care for the whole person" or when the relational attitude in spiritual care is emphasized (Hölsgens 2020; Visser et al. 2022). However, care professionals do not always recognize the experiences and questions that show spiritual needs (Hölsgens 2020; Smeets 2022). Recent interdisciplinary initiatives do indicate that, when care professionals have come to know chaplains, they have clearly seen differences in their expertise and approaches. Oftentimes, this is framed as a difference in primary and secondary orientation in care encounters, where the primary orientation of care professionals is said to be on the physical, emotional, and/or social domains, and their secondary orientation is on the spiritual domain. For chaplains, this is said to be the other way around (Mooren 1989; Toom 2022; Zock 2022a). In our current study, we are additionally finding that professionals collaborating with chaplains see a difference in ways of working, in which chaplaincy encounters tend to be longer and less solution-oriented than care encounters by other professionals. However, a recurring observation is that direct experience with chaplains seems to be essential for these differences in orientation and approach to become clear to professionals. This relates back to professional identity being implicit or embodied, rather than explicit (Ellis and Hogard 2020; Muthert et al. 2019).

Some chaplains suggest that a contrast between "daily meaning" and "existential meaning" might enhance the external spiritual identity of chaplaincy (Jacobs 2020). Daily meaning refers to experiences of meaning in everyday situations, i.e., what makes it worthwhile to get out of bed each morning and to engage in daily activities? (Smaling and Alma 2010, p. 17). This type of meaning is more related to meaning *in* life and is closer to the practice of all care professionals. It might also be more easily subject to change, lending itself more readily to short and focused interventions. Existential meaning refers to meaning in relation to life in general, i.e., what constitutes a "good" life and what is my role in this? (Smaling and Alma 2010, p. 17). This type of meaning is more closely related to meaning *of* life and often surfaces in crisis situations. Questions around the meaning of life are also more closely related to religious and worldview traditions. Such questions are often referred to as "slow questions" or "ultimate concerns" that are not readily answered, if ever an answer is found.

At first sight, the contrast between daily meaning and existential meaning might be a promising avenue to use to clarify the external professional identity of chaplaincy in a pluralistic and multidisciplinary context. The term "existential meaning" bears a less instrumental and intentional, and a more religious or spiritual, connotation. Zock (2022b) notes that chaplaincy is often associated with the existential, both in the Netherlands and elsewhere. Nevertheless, she observes that in most of the Dutch writing about chaplaincy, the existential domain is distinguished from the spiritual and religious. In Dutch, the spiritual domain is more often connected to transcendence whereas the existential domain lies in "everyday reality". Religion is most often used to refer to institutionalized worldviews that include a notion of God. Thus, the term existential would not fully resolve the concerns around the marginalization of religion. Also, "the everyday" and "the existential" are not separate dimensions; our sense of meaning of life shows itself in how we find meaning in life, and our experience of meaning in life affects how we understand the meaning of life

(Park 2010). Finally, we agree with Smeets (2020) and Alma (2020) that meaning happens in everyday life (which, of course, is not a new idea, see Haardt and Korte 2002, for example). Thus, the chaplaincy encounter also focuses on the daily life of the client. So, in practice, it remains unclear where to draw the line between the responsibilities of chaplains and of other care professionals. It is also unclear to what extent care professionals are familiar with the terminology of "daily meaning" and "existential meaning" and how they relate to this terminology. Finally, these terms retain the cognitive connotation of "meaning".

We conclude, therefore, that the move to meaning does not seem to substantially clarify the professional identity of chaplaincy in the Netherlands. The chaplains critiquing the move instead plead for an understanding of chaplaincy and multi- or interdisciplinary spiritual care that

a.   Pays more attention to the pluralistic and everyday manifestations of religion, spirituality, and meaning;
b.   Stays closer to the internal professional identity of chaplaincy as rooted in religious/spiritual life views and a relational attitude;
c.   Helps to navigate professional boundaries.

We suggest that putting hermeneutical practice back at the center of chaplaincy will assist with this.

## 4. Perspective—Hermeneutic Competency

In the professional standard by the Dutch Association of Spiritual Caregivers (2015), hermeneutic competency is defined as: "being able to clarify questions about meaning and beliefs and habits in relation to the context or situation, and being able to offer worldview counselling. This includes the ability to understand, interpret, and translate meaning in texts and images, practices and life stories, traditions and new forms of meaning-making. This takes place in relation to and in exchange with existential and spiritual questions, philosophical and ethical sources, contemporary society, faith and culture. Crucial is the ability to hear and clarify emotions and unspoken questions and implicit assumptions." What this competency asks of the chaplain is to carefully listen to, look at, and feel what is being expressed in an encounter, to negotiate different ways of understanding what is being expressed, and to articulate this in a way that matches the situation but might shed a different light on it. Such skills ask for an integration between skills and theories from the humanities, where human narratives are viewed as lived, contextual, and dynamic constructions, and those of the psychosocial sciences, where the focus is more on personal and individual one-to-one exchanges.

This competency is often seen as the core competency of chaplains. The impression, however, is that in education and discourse around chaplaincy, it has become implicit rather than explicit (Muthert et al. 2019). And perhaps chaplains are losing this competency somewhat, given the observation that they seem to rely often on the strategy of neutralizing in their communication amongst each other and with other professionals. In interfaith chaplaincy encounters, chaplains also tend to use neutralizing as a tactic (Cadge and Sigalow 2013; Liefbroer and Olsman 2020).

Cadge and Sigalow contrast neutralizing with code-switching, which they define as using "the languages, rituals, and practices of the people with whom they work". Grefe and McCarroll (2022) identify code-switching as one of the core skills of chaplains in the 21st century (interestingly, this skill is included under "meaning-making competencies"). In our view, hermeneutic competency goes further than code-switching in that it does not only ask of the chaplain to "speak the language of the other" but to more broadly acknowledge, discuss, learn from, and/or facilitate different faiths and disciplinary approaches to spiritual care in relation to one's own faith and disciplinary approach (Walton 2021; Smeets 2020). It is about a relational engagement in which all parties play an active role and negotiate the meanings which are expressed. Code-switching is more one-sided, where one party adopts the cultural expressions of the other, but this is not necessarily a reciprocal process (Morton 2014). In some sense, neutralizing and code-switching can go hand-in-hand when

language of another is used to try to emphasize commonalities. This also relates back to the critique in Section 3 regarding the instrumentalization of chaplaincy for external goals.

We want to stress here that we do not propose reintroducing hermeneutics in exactly the same way as it was practiced in the past, but—instead—to use what we have learned in the past and adjust it to the demands of the current context. Putting this revitalized competency back at the center of chaplaincy, we believe, creates space for the pluralistic and everyday manifestations of religion, spirituality, and meaning; it invokes the core of chaplaincy; and it allows for more open exploration of and diversity in professional boundaries.

In this revitalized process of hermeneutic engagement, it is likely that a more common language and understanding of interdisciplinary spiritual care will develop, given that this is needed to effectively communicate and collaborate. The difference from the current situation will be, however, that this will develop in a more context-specific and bottom-up process, which allows for more careful negation of concepts, understandings, and differences than the current nation-wide approach. Thus, we expect that this prevents the understanding of one of the disciplines becoming the dominant frame, even when terminology is chosen that is more prevalent in one of the disciplines (Bloomfield et al. 2020). We are seeing this work. For example, in a national study on the organization of chaplaincy in primary care and the social domain, it was found that chaplaincy organizations saw more clients and seemed more financially sustainable when they collaborated more closely with other professionals through adjusting their PR, acquisition strategies, and services to match their chaplaincy expertise, how the other professionals viewed the spiritual needs of their clients, and the types of spiritual care services the professionals needed (Visser et al. 2022).

Revitalizing the hermeneutic competency of chaplains will pose some challenges to chaplaincy education. It asks of chaplaincy educators that they not only teach students to use their hermeneutical skills in client encounters, but also in interdisciplinary encounters. In this regard, Bloomfield et al. (2020) suggest incorporating more interprofessional education in care curricula. In such education, students from different disciplines collaborate with each other on real-life scenarios to learn more about each other's professions and how to communicate effectively with each other. Several Dutch (and other) universities are currently developing training to this end, both in initial education of chaplains and for in-company training.

**Author Contributions:** Conceptualization, A.V. and H.Z.; investigation, A.V., H.M. and H.Z.; writing—original draft preparation, A.V.; writing—review and editing, H.M. and H.Z. All authors have read and agreed to the published version of the manuscript.

**Funding:** This research received no external funding.

**Conflicts of Interest:** The authors declare no conflict of interest.

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
