# Peer review of "Positioning Chaplaincy in the Pluralistic and Multidisciplinary Dutch Care Context"

_religions, doi:10.3390/rel14091173_

Round 1
Reviewer 1 Report
The article is quite pertinent, because although it develops in the context of the Dutch debate on the identity of chaplaincy, it can shed light on other contexts. The proposal of code-switching seems to be an interesting strategy for different linguistic and conceptual contexts and between members of a multidisciplinary team, with different backgrounds and theoretical orientations of the subject.
I would also like to point out some questions that arise when reading the text, in order to help the author to think about such dimensions around the theme:
1. Because it is a complex phenomenon, the issue of spirituality and health requires such communication skills, and code-switching, as I said above, is a good theoretical tool that facilitates and even enables interaction between plural perspectives of analysis and operationalisation of concepts. However, at some point it is necessary to carry out an exercise of shared translation of concepts to be operationalised among the plurality of actors involved in a multidisciplinary team. Such a moment is fundamental for the evolution of a multidisciplinary team towards interdisciplinary and even transdisciplinary work, whose basic element is the shared use of concepts, translated into the different epistemologies involved in the team. Such translation is an important step so that there is no dominant area donating the concepts, but even if such a concept is emergent from one area, its joint translation allows a more adequate epistemological and practical appropriation between the areas. It would be interesting if the author said something about code-switching and interdisciplinarity or transdisciplinarity, in the sense of thinking about the role of shared concepts between different areas, and not only decoding concepts, as a previous moment. In this sense, wouldn't code-switching be a means to qualify the debate between the main trends rather than an end?
2. Discourse analysis also implies realising which social practices are involved in each discourse. Divergence often has political (or ideological) reasons within a community and not necessarily logical ones. Sometimes discursive practices are distinct, but the social practices involved in them are convergent. Are there convergent elements regarding spiritual care practices that remain between the dissonant perspectives listed by the author?
3. The author mentions code-switching works applied to chaplancy work, but it would be interesting to at least point to what are the main theoretical influences of code-switching that have been used in these works in the Dutch context. There are several code-switching authors such as John Gumperz, Carol Myers-Scotton, Penelope Eckert and Sally McConnell-Ginet, Peter Auer, Kathryn Woolard, Monica Heller, Lesley Milroy and others. But no mention of which theoretical trend is installed in chaplaincy work.
It is expected that the author(s) will take a position on such issues, either to develop them or to point out as tasks to be accomplished.
Congratulations on the work and the proposal to advance the debate!
Author Response
Thank you for your positive response to our perspective and your questions. They have challenged us to think the issue through further and sharpen our view. This has led to a revision of our idea, focusing on hermeneutical skill rather than code-switching. Though these two are related, code-switching indeed describes the issue at hand but not necessarily its solution. Some form of mutual understanding will indeed be necessary and code-switching might not optimally facilitate that. Understanding differences and navigating them requires hermeneutical skill.
We did not know how address the second question asked within our manuscript, but we can try to provide an answer here. There seems to be consensus in the field about the three problems mentioned, but also that meaning is an important element in chaplaincy and spiritual care. The difference lies in the weight that is attached to the problems. Some people don't see them as large threats to the field and see potential in moving on with meaning as the central concept. Others see them as the demise of chaplaincy.
We have now given less prominence to code-switching in the discussion, so felt it would not be fitting to include a discussion of its background in chaplaincy discourse, but to briefly answer the reviewer's question: Thought about code-switching is hardly present in Dutch chaplaincy. The concept was introduced in 2017 by Liefbroer and was briefly picked up by two authors, but has not been brought further in the field.
Reviewer 2 Report
line 61 - how a professional (SINGULAR) works and why they (PLURAL) do so in this way.
I would have included at least a paragraph or two on trends in neighbouring countries or the US or elsewhere, simply for contrast purposes, but I understand the author's sharp focus on his topic.
Author Response
Thank you for your positive response to the perspective. We have corrected the grammatical error in line 61. We agree with the reviewer that a paragraph describing trends in the neighboring countries and the US would have given context to the discussion, the Netherlands is certainly not the only country in which the professional identity of chaplains is questioned and in which similar discussions play out. They are also present in the US, Switzerland, Denmark, Norway and probably elsewhere as well. However, we indeed felt that it would have distracted from the topic and we also have the impression that the discussion is sharper in the Netherlands than elsewhere, because the questions have been plaguing us for about 50 years already. We thank the reviewer for understanding our choice.